# Modified Orange Peel Waste as a Sustainable Material for Adsorption of Contaminants

**DOI:** 10.3390/ma16031092

**Published:** 2023-01-27

**Authors:** Uloaku Michael-Igolima, Samuel J. Abbey, Augustine O. Ifelebuegu, Eyo U. Eyo

**Affiliations:** 1Department of Geography and Environmental Management, Faculty of Environment and Technology, University of the West of England, Bristol BS16 1QY, UK; 2Office of the Deputy Vice Chancellor Academic Affairs, Victoria University, Kampala P.O. Box 30866, Uganda

**Keywords:** adsorption, biosorption, biomass, orange peel, structural properties

## Abstract

World orange production is estimated at 60 million tons per annum, while the annual production of orange peel waste is 32 million tons. According to available data, the adsorption capacity of orange peel ranges from 3 mg/g to 5 mg/g, while their water uptake is lower than 1 mg/g. The low water uptake of orange peel and the abundance of biomass in nature has made orange peel an excellent biosorption material. This review summarised different studies on orange peel adsorption of various contaminants to identify properties of orange peel that influence the adsorption of contaminants. Most of the literature reviewed studied orange peel adsorption of heavy metals, followed by studies on the adsorption of dyes, while few studies have investigated adsorption of oil by orange peel. FTIR spectra analysis and SEM micrographs of raw and activated orange peels were studied to understand the structural properties of the biomass responsible for adsorption. The study identified pectin, hydroxyl, carbonyl, carboxyl, and amine groups as components and important functional groups responsible for adsorption in orange peel. Furthermore, changes were observed in the structural properties of the peel after undergoing various modifications. Physical modification increased the surface area for binding and the adsorption of contaminants, while chemical treatments increased the carboxylic groups enhancing adsorption and the binding of contaminants. In addition, heating orange peel during the thermal modification process resulted in a highly porous structure and a subsequent increase in adsorption capacities. In conclusion, physical, chemical, and thermal treatments improve the structural properties of orange peel, resulting in high biosorption uptake. However, orange peels treated with chemicals recorded the highest contaminants adsorption capacities.

## 1. Introduction

Adsorption is a simple, low cost, and eco-friendly physical remediation method. Different adsorbents, including synthetic, natural inorganic, and natural organic sorbents, have been applied in removing various contaminants from environmental media [1,2]. Synthetic adsorbents are the most extensively used sorbents; studies have shown they have high hydrophobic and oleophilic properties and can sorb up to 70 times their weight in water. Furthermore, they can be used several times after recovery; however, they are expensive and are usually nondegradable [3]. Natural inorganic adsorbents have low sorption capacities 4–20 times the weight of the adsorbent with little buoyancy, thus they have less application in environmental clean-up [2,4,5]. Conversely, natural organic adsorbents are carbon-based biomass materials derived from plants or animals and are mostly found in the form of agricultural wastes. They are made from renewable sources that are non-toxic, noncorrosive, fully active upon recycling, and cost efficient. Recently, there has been renewed interest in natural organic adsorbents, low-cost non-conventional biomass from agricultural wastes, such as cotton fibres, rice husks, and kapok fibres, including various fruit wastes, such as bananas, and orange peels have been applied in environmental clean-up [6,7,8], as shown in Table 1.

Studies have shown that agricultural wastes have great potential to be used as adsorbent material due to the huge biomass resource and contribution to the prevention of global warming. About 5 billion metric tonnes of agricultural waste are annually produced worldwide, the equivalent of about 1.2 billion tonnes of oil, or 25% of current global production. The annual production of orange peel is estimated to be 32 million tonnes [9], making orange peel an available material for the biosorption of contaminants. Extensive studies have been carried out on the application of orange peel in environmental clean-up; however, most of the literature features studies on the adsorption of heavy metals and dyes [10,11,12], while very few studies have reported on the oil adsorption potential of orange peel [13,14]. Furthermore, available data revealed that the adsorption capacities of orange peel range from 3 mg/g to 5 mg/g, while the water uptake is lower than 1 mg/g. Although the adsorption capacity of orange peel is considered low compared to kapok fibre and cotton fibre, the low water adsorption capacity and abundance of orange peel in nature makes it an excellent biosorption material for various contaminants [15,16,17,18]. Therefore, the aim of this study is to review several instances in the existing literature on the adsorption of various contaminants by orange peel and identify structural properties of the peel that influence contaminants adsorption. In achieving this, SEM images and FTIR spectra of raw and modified orange peels would be extensively studied to determine properties of the peel that influence biosorption.

**Table 1 materials-16-01092-t001:** Summary of studies on contaminants removal by natural organic adsorbents.

S/No	Sorbents	Adsorbent Base	Adsorbent State	Adsorption Capacities (mg/g)	References
1	Banana peel	Cellulose	Modified	32.40 68.92 99.09	[19]
2	Sugar beet pulp	Cellulose	Modified	73.53	[20]
3	Rice husk	Cellulose	Modified	6.0–9.0	[21]
4	Vegetable fibres	Cellulose	Raw	85.0	[7]
5	Oil Palm empty fruit bunch	Cellulose	Modified	7.0	[22]
6	*Posidonia oceanica* (L.)	Cellulose	Raw + Modified	4.74 12.80	[23]
7	Sugarcane leaves straw Sugarcane bagasse	Cellulose	Raw	8.0 6.6	[24]
8	Green macroalgae	Cellulose	Modified	19.38–23.08	[25]
9	Coconut shell	Cellulose	Modified	2.48	[26]
10	Coconut fibre	Cellulose	Modified	13.2–14.0	[27]
11	Corn cob	Cellulose	Modified	4.21–7.80	[28]
12	Wheat straw	Cellulose	Modified	41.84	[29]
13	Pineapple leaf waste	Cellulose	Modified	37.9	[30]
14	Wheat bran	Cellulose	Modified	62	[31]
15	Hazelnut shells	Cellulose	Modified	41.3	[32]
16	Papaya seed	Cellulose	Modified	55.6 37.43	[33]
17	Sunflower stalk	Cellulose	Modified	39	[34]
18	Chicken feathers	Keratin	Modified	6.1	[1]
19	Human hair	Keratin	Raw + Modified	8.1 5.5	[3]
20	Mango peel	Cellulose	Modified	46.09 39.75 28.21	[35]
21	Palm ash	Cellulose	Modified	61	[36]
22	Palm shell	Cellulose	Modified	83.33	[37]
23	Sunflower stalk	Cellulose	Modified	182.90 69.80	[38]
24	Orange peel	Cellulose	Modified	200	[39]
25	Rice husk	Cellulose	Modified	15.0	[40]
26	Cashew nutshell	Cellulose	Modified	22.11	[41]
27	Cotton fibre	Cellulose	Modified	25–75	[42]
28	Spider cuticles	Keratin	Raw	12.6–16.6	[43]
29	Sugarcane bagasse	Cellulose	Modified	38.03	[44]
30	Bamboo leaf powder	Cellulose	Modified	28.1	[45]
31	Wastepaper	Cellulose	Modified	24.4	[46]
32	Silk fibre	Cellulose	Modified	46.83	[47]
33	Paper waste	Cellulose	Modified	29.67	[48]
34	Hazel nutshell	Cellulose	Modified	28.18	[49]
35	Wool fibre	Keratin	Raw + Modified	12.0	[50]
36	Wool fibre	Keratin	Raw	11.06	[51]
37	Grapefruits	Cellulose	Modified	37.43 39.06	[52]
38	Chicken feathers	Keratin	Modified	50.0	[53]
39	Garlic peel and Onion peel	Cellulose	Modified	3.85 4.55	[54]
40	Pine leaf powder	Cellulose	Modified	3.27	[55]
41	Banana stalk	Cellulose	Modified	138	[56]
42	Wheat straw	Cellulose	Modified	6.91	[57]
43	Banana	Cellulose	Modified	5	[58]
44	Rice husk	Cellulose	Modified	6.22	[59]
45	Rice husk	Cellulose	Modified	19.66	[60]
46	Kapok fibre	Cellulose	Modified	46.9–58.8	[61]
47	Corn stalk	Cellulose	Modified	21.37	[62]
48	Pigeon feathers	Keratin	Modified	30.0	[63]
49	Sugarcane bagasse	Cellulose	Modified	13.72	[64]

## 2. Adsorbent Materials

Adsorption materials, also known as adsorbents, are materials that can effectively adsorb substances. The existing literature shows that a wide range of materials have been extensively studied as adsorbents [3]. They are usually grouped into three categories, including synthetic organic, natural inorganic, and natural organic materials. Synthetic organic materials, such as polyacrylate, polystyrene, polypropylene, and polyurethane, are the most extensively used adsorbents. They have high adsorption capacities and are commercially available; however, they are expensive and non-degradable. Natural inorganic substances, such as clay, zeolites, and bone char, are easily available in nature, making them attractive sorbent materials for contaminants removal. Nonetheless, the adsorption capacities of most natural inorganic materials are low. While natural organic adsorbents are cheap, recyclable, and abundant in nature, making them excellent biosorption materials for the removal of contaminants; examples include kapok, cotton, fruit peels, and rice husks [1,3].

### 2.1. Characteristics of Good Adsorbent Material

A direct expression of surface wettability is the contact angle (CA). Surfaces with high water contact angle of ≥150° and a minimal value of contact angle hysteresis are rough surfaces and are usually referred to as super-adsorbents [65]. The roughness of a surface enhances both the hydrophobic and oleophilic nature of a material, which confirms Wenzel’s theory on wettability that the contact angle increases with porosity and the surface roughness of a material. Plants such as lotus leaves, cotton fibres, and kapok fibres with naturally occurring superhydrophobic surfaces have high water contact angle. Lotus leaves have a water contact angle (CA) of 161°, and a structure that consists of a combination of two scale roughness of 10 µm (rough structure) and a 100 nm fine structure. The hydrophobic nature of lotus leaves arises from a combination of the epicuticular wax secreted from the leaf itself and the roughness. Kapok fibre has a water contact angle (CA) that ranges from 138.6° to 151.2° depending on the location, making it a superhydrophobic surface [66]. Some researchers have reported that kapok fibre is mainly composed of cellulose, lignin, and pentosan [67], while others have opined that kapok fibre comprises of cellulose, lignin, xylene, and high levels of acetyl groups [68,69]. Kapok fibre possess waxy coatings and a large hollow structure that gives it a porosity of 80%, which is responsible for the hydrophobic nature of kapok leaves. According to studies, the water contact angle of cotton fibre is 100°, cotton fibre is about 90% cellulose, and the non-cellulose part includes proteins, waxes, pectin, inorganics, and other substances; however, the composition of cotton fibre differs based on location and maturity [70]. The water contact angle of orange peel is reported at 0°, which is responsible for the low sorption capacity of orange peel compared to lotus leaves, kapok, and cotton fibres [14]. Table 2 shows the water contact angle of some high sorption lignocellulose materials and their adsorption capacities.

### 2.2. Chemical Composition and Physical Properties of Orange Peel

The chemical composition of orange peel differs based on location, varieties, level of maturity, and growing conditions [73]. According to Mafra [74] orange peel is 97.83% organic matter and contains carbon, hydrogen, oxygen, nitrogen, sulphur, chloride, and ash. Bampidis [75] further confirmed the organic matter composition of orange peel, reporting that the dry matter of orange peel is mainly organic matter containing proteins and short-chain organic acids no more than four carbons. Others opined that orange peel contains soluble sugar, starch, and fibre, including cellulose, hemicellulose, lignin, and pectin, as well as ash, fat, protein, and about 1% organic acids [76]. The chemical composition of orange peel, as summarized in Table 3, shows that the peel mainly consists of organic matter.

In addition, several studies on the physical properties of orange peel have revealed that it contains cellulose, hemicellulose, lignin, and pectin [13,74]. Cellulose is the most abundant polysaccharide found in the cell walls of plant biomass. The chemical formula of cellulose is (C_6_H_10_O_5_) n, where n represents the number of glucose groups, ranging from hundreds to thousands. Cellulose is insoluble in water, dilute acid, and dilute alkali solutions, and is recalcitrant to hydrolyses and enzymatic activities due to their strong hydrogen bonds [79]; the chemical structure of cellulose is presented in Figure 1.

Unlike cellulose, hemicellulose has a random, amorphous structure composed of various sugar monomers, which have little physical and chemical resistance, and their polymerization degree is between 50–200 °C. Hemicelluloses are mostly soluble in water and dilute in alkali solutions at temperatures above 180 °C, but they are insoluble in water at temperatures below 180 °C and account for about one third of total biomass weight [80,81]. Together with cellulose and hemicellulose, lignin is found in the cell walls of plants; it is the second most abundant compound of plant biomass with a complex hydrocarbon polymer that consists of both aliphatic and aromatic compounds. The basic monomeric units of lignin are P-hydroxyphenyl, guaiacyl, and syringyl. Lignin is hydrophobic in nature, totally insoluble in most solvents, and is thermally stable but prone to UV degradation [82,83]. Pectin is a complex polysaccharide found in the cell walls of plants and contributes to the firmness and structure of plant tissues. They are composed of 1,4 α-D-galacturonic acid (GA) in free or esterified form. The basic monomer of pectin includes rhamnose, galactose, arabinose, glucose, and xylose [84].

## 3. Testing Techniques and Results

Fresh orange peels (OP) were locally sourced from a smoothie shop in Bristol. The peels were washed with water to remove impurities and dried in an oven at 75 °C for 48 h. The dried orange peel was ground and sieved to a powder and was used for different orange peel modification processes. The different stages of orange peel preparation are presented in Figure 2.

### 3.1. Adsorption Properties of Orange Peel

The different FTIR spectra analysis of orange peel studied in this paper identified pectin, hydroxyl, carbonyl, carboxylic, and amine groups as important components and functional groups responsible for adsorption in orange peel [13,58,77]. The FTIR spectrum of raw orange peel in Figure 3a displays several peaks, demonstrating the complex nature of orange peel. The intense signal at 3290.87 cm^−1^ is attributed to the presence of hydroxyl groups usually present in carbohydrates and phenolic groups in fruit peels; the peak at 2921.63 cm^−1^ could indicate the asymmetric and symmetric stretching of C-H, which corresponds to the FTIR spectra in Zapata [76] and the band at 1737.24 cm^−1^ is assigned to carbonyl groups present in the ester bond, while the band at 1606.34 cm^−1^ is attributed to the aliphatic and unsaturated aromatic compounds. The band at 1428 cm^−1^ is assigned to the C-H bending vibration, which forms the basic structure of lignocellulosic material, and the stretching at 1011.57 cm^−1^ could indicate the presence of C-OH and C-OR groups present in carbohydrates [13,77].

The SEM image of raw orange peel in Figure 3b shows a clean, smooth, and non-porous structure. The components of orange peel, including cellulose, hemicellulose, lignin, and pectin, are the reason for the smooth surface structure. Smooth surfaces have low adsorption uptake due to the reduced number of active sites available for the adsorption of contaminants, and do not allow significant binding of contaminants, hence the reason for the low biosorption uptake of orange peel [13,14]. Roughness and porosity are usually linked to the high sorption capacities of biomass because they provide more active sites for adsorption than smooth surfaces. However, studies have shown that the adsorption capacities of a biomass can be improved by increasing the surface roughness and porosity using different modification methods, including physical, chemical, and thermal methods. Hence, the aim of the modification process is to improve the structural properties of the biomass, resulting in an increased binding site and biosorption uptake of the biomass [13,14,74].

### 3.2. Physical Modification

Physical modification involves washing, drying, grinding, and crushing orange peels to increase the surface area and adsorption capacity of the peel [85,86]. In this study, fresh orange peels were washed, dried at 75 °C for 48 h, crushed, and sieved to a powder. The FTIR spectrum of the physically modified orange peel, shown in Figure 4a, identified several bands different from the bands in the raw orange peel spectrum shown in Figure 3a. The SEM image of the characterised orange peel shown in Figure 5c reveals a heterogenous and porous structure, indicating the presence of an effective structural morphology for the adsorption of contaminants. Furthermore, Mafra [74] studied the effect of physical characterisation of orange peel on dye adsorption. In their study, crushed orange peel powder was used as an adsorbent to remove dye (Remazol Brilliant Blue R) in organic wastes, and the study reported a maximum adsorption uptake of 11.62 mg/g. The SEM micrograph from their study, shown in Figure 5a, revealed a heterogenous porous structure, which corresponds to the image obtained in this study, as shown in Figure 5c. The SEM image of the characterised orange peel after adsorption reveals that pore spaces within the peel were filled with contaminants, confirming the highly porous structure of the orange peel after physical characterisation [16,85]. Kumar [87] also reported the effect of particle size on the adsorption capacities of orange peel, confirming that the lower the particle size, the higher the adsorption capacity.

### 3.3. Chemical Modification

Incorporating chemicals into biomass improves the structural properties of the biomass and increases the adsorption capacity. The effects of different chemicals on the structural properties and adsorption capacities of orange peel have been studied by many researchers [91,92]. Marin [91] studied the role of three major functional groups (amine, carboxyl, and hydroxyl) found in orange peels on the adsorption of heavy metals using chemically modified (esterification, methylation, and acetylation) orange peels. The study confirmed the modification of carboxyl groups and the extraction of pectin in the esterified orange peel. The esterification treatment blocked the COOH groups reducing the binding of metals, which resulted in a decreased biosorption capacity of the orange peel. Adsorption studies showed low biosorption uptake for the esterified biomass due to the destruction of carboxyl groups and the extraction of pectin during the esterification process, which confirm reports that carboxyl groups and pectin are the major functional groups and components responsible for adsorption in fruit peels [14,17,91]. The acetylated and methylated sorbents did not show significant differences in terms of structure and biosorption uptake, suggesting that amine and hydroxyl groups have little effect on adsorption in orange peel. Khalfaoui [18] also confirmed the low biosorption uptake of acetylated and methylated orange peels; they further reported that caustic soda, sulphuric acid, and methanol-modified orange peel showed the highest biosorption uptake, confirming reports that incorporating sulphuric acid into biomass results in high biosorption uptake [93,94].

According to several studies, treating orange peel with a base increases the carboxylates and binding sites of the peel [88,95,96,97]. Orange peel components, including cellulose, hemicellulose, and lignin, contain methyl esters, which do not allow significant binding of contaminants. However, methyl esters can be converted to carboxylates by treating orange peel with a base, such as sodium hydroxide. In this study, 150 g of dried orange peel powder was soaked in a mixture of 500 L of NaOH and 500 L of ethanol. After 24 h, the mixture was washed with distilled water until PH7 was reached, and the obtained OP was dried in an oven for 48 h. The FTIR spectra of characterised orange peel in Figure 4b shows several adsorption bands; however, the weak band at 1605.64 cm^−1^ suggests that methyl ester is hydrolysed and converted to carboxylates by sodium hydroxide. In a series of adsorption experiments, Feng [35,36,37] studied the effects of sodium hydroxide on the orange peel adsorption of metals. Feng [95] increased the carboxyl groups in orange peel by incorporating sodium hydroxide and calcium chloride into orange peel, resulting in the increased biosorption uptake of contaminants. The FTIR spectrum in their study, shown in Figure 4e, reveals a weak band at 1744 cm^−1^ which corresponds to the spectrum in Figure 4b, and the study reported a high adsorption capacity of 289 mg/g. Furthermore, Sha [89] studied the adsorption capacities of mercapto-acetic-acid-modified orange peel; the FTIR spectra of the modified orange peel before and after adsorption in Figure 4g show some distinct changes. The weak band at 667 cm^−1^ suggests the presence of sulphur groups, while the shift in the band at 3417 cm^−1^ after adsorption indicates the involvement of hydroxyl groups in the adsorption, which corresponds to the spectra in Liang [90] as shown in Figure 4h. In addition, the weak carbonyl band at 1735 cm^−1^ and the C=O band at 1630 cm^−1^ suggest some carbonyl binding. Other authors have reported similar changes in the properties and structure of orange peel after treatment with NaOH [10,94]. Furthermore, the SEM image of NaOH-modified orange peel in this study reveals highly hierarchical and porous structure evidence of a good sorbent.

### 3.4. Thermal Modification

Furthermore, several studies have shown that heating orange peel at high temperatures improves the properties and surface structure of the peel [13,40,97,98,99,100]. Dried orange peels heated at 300 °C to 500 °C had increased surface area; however, a decrease in surface area was reported at 550 °C, indicating that heating citrus peels at extreme high temperatures reduces porosity and surface area of the biomass. The FTIR spectra in Figure 4c,d show images of calcined (300 °C and 500 °C) orange peels from this study. The peak at 29,310 cm^−1^ in the raw spectrum, usually attributed to the aliphatic groups, is reduced in the 300 °C spectrum, which corresponds to the image in Khalfaoui [101] as shown in Figure 6B, and disappears at 500 °C; while the band at 1639 cm^−1^, indicating the presence of an amine group, is reduced at 500 °C. The SEM micrographs of orange peel calcined at 300 °C to 500 °C show very porous structures, characteristics of a good adsorbent material. An increase in adsorption capacities has also been reported when a calcination temperature of 300 °C was increased from 30 min to 1 h, while an increase to 2 h did not increase adsorption capacities. The same was reported for calcination temperatures at 400 °C, as shown in Figure 6e,f. However, at a calcination temperature of 550 °C, a decrease in adsorption capacities was reported when calcination time was increased to 2 h, as shown in Figure 6g. El-Gheriany [13] also studied the effects of pyrolysis on the biosorption of orange peel; the SEM micrographs of the pyrolyzed orange peels, shown in Figure 5d, reveal rough and porous structures after heating the peel at 500 °C, which compares to the SEM image reported in this study. The presence of pores was attributed to the release of gases during the heating process. The highest adsorption uptake was reported at 500 °C, and the increase in adsorption was attributed to the increase in pore volume in the pyrolyzed biomass. The study further observed that the biosorption uptake of orange peel declined when orange peel was heated above 500 °C, which confirms the report of Khalfaoui [101] shown in Figure 6D. Furthermore, Zapata [76]) observed a mass decrease in samples during the thermal process, which continued to 580 °C; the decrease was attributed to the dehydration and thermal degradation of the sample. Below 100 °C was attributed to the release of weak bonds of water, while the mass decrease of the sample from 200 °C to 580 °C was attributed to the decomposition of hemicellulose, cellulose, and lignin, the main components of orange peels.

### 3.5. Thermochemical and Physical Modifications

The physical characterisation of biomass increases the surface area, while chemical treatments increase the number of binding sites, and the heating of biomass during thermal treatment increases the pore spaces within a biomass. Several researchers have reported that combining chemical and thermal treatments is a more efficient process for enhancing the adsorption capacities of a biomass [10,17,18]. In a study conducted by Abid [19] orange peels were dried and crushed to a particle size lower than 250 µm to increase the surface area of the peel. The particles were treated with H_2_SO_4_, washed with deionised water until PH7 was obtained, and then dried for 48 h at 65 °C to obtain charred orange peel adsorbent, with a high adsorption capacity of 60.9 mg/g. Furthermore, Khaled [17] reported a high sorption uptake of 107.53 mg/g following the physical characterisation and thermochemical modification of orange peel. FTIR spectrum analysis of the functionalized orange peel showed a band around 3480 cm^−1^ attributed to OH stretching vibration, and a lower peak intensity linked to the presence of H_2_SO_4_. The weak band around 3835 cm^−1^ was linked to aliphatic groups stretching vibration, while the band at 1620 cm^−1^ was linked to C=C stretching in aromatics due to the carbonisation of the orange peel. New adsorption bands were seen at 1383 cm^−1^ and 580 cm^1^, which were attributed to the asymmetric and symmetric stretching of SO_2_ and S-O groups. In another study, Khalfaoui [18] followed the same thermochemical process as Khaled [17] to prepare superhydrophobic oleophilic adsorbents from orange peel. The adsorption rate and capacity were faster and higher in the super adsorbent than the untreated orange peel, suggesting that the functional groups responsible for adsorption in orange peel are favourably affected by the three modification processes of physical, chemical, and thermal treatments. Table 4 is a summary of studies on orange peel adsorption of contaminants.

## 4. Discussion

The exact chemical composition of orange peel varies, as with all plants and fruits, based on location, growing conditions, maturity, and variety. Orange peel is mainly organic matter, containing proteins and other short-chain organic acids, no more than four carbons, as shown in Table 3. The SEM micrograph of raw orange peel shown in Figure 3b reveals a clean and smooth surface structure composed of cellulose, hemicellulose, pectin, and lignin. Existing studies have shown that smooth surfaces have low adsorption capacities compared to rough and porous structures, which have more active sites for the adsorption of contaminants [13,14]. However, the adsorption capacities of orange peel can be improved by increasing the roughness and porosity of the peel using physical, chemical, and thermal modification processes; the adsorption capacities of various orange peel modifications reviewed in this study are summarized in Table 4. SEM micrographs of the modified orange peels showed huge differences in the microstructure and the morphology of raw and treated orange peels. The physical characterisation of orange peel increases the surface area, and subsequently increases the rate of contaminants adsorption. Adsorbents with finer particles have large surface areas and high sorption capacities [87]. In addition, heating orange peel at high temperatures increases the porosity and surface area; however, studies have shown that heating orange peel at high temperatures above 500 °C for a longer duration of up to 2 h reduces the porosity and the surface area of the biomass, resulting in reduced contaminant adsorption [13,101]. Incorporating chemicals into orange peel enhances structural properties, such as porosity and roughness of the peel, which results in increased biosorption capacities of the peel. Orange peel components, including cellulose, hemicellulose, and lignin, contain methyl esters, which do not allow significant binding of contaminants. Treating orange peel with chemicals, including bases like sodium hydroxide, converts methyl esters to carboxylates, thereby increasing the number of carboxylates and the adsorption capacities of the peel.

Furthermore, the adsorption capacities of modified orange peels reviewed in this study, as shown in Figure 7, reveal that orange peels treated with chemicals recorded the highest contaminant adsorption capacities. Several studies have reported that carboxylic groups and pectin are the major functional groups and components responsible for contaminants adsorption in orange peel. However, incorporating chemicals into orange peel increases the carboxylic groups, resulting in increased biosorption capacities of the peel [95,104]. In addition, combining chemical and thermal methods is an efficient method for increasing the adsorption capacities of orange peel. SEM micrographs of thermochemical modified orange peel in Abid [19] revealed large surface areas, rough structures, and an increased number of pore spaces within the peel, which resulted in the increased biosorption capacities of the orange peel. Additionally, Khaled [17] reported high biosorption capacities of 107.53 mg/g following the combined treatment of orange peel. Orange peels modified through thermal and physical treatments showed increased biosorption capacities when compared to the biosorption capacities of raw orange peel, which ranged from 3 mg/g to 5 mg/g.

## 5. Conclusions

The study reviewed different instances in the literature on the adsorption of contaminants by orange peel. FTIR spectra analysis of orange peel identified pectin as the main component responsible for adsorption in orange peel, while hydroxyl, carboxyl, carbonyl, and amine groups are important functional groups responsible for adsorption in orange peel. The SEM micrograph of raw orange peel revealed smooth and clean surface structures, identified as the key reason for the low biosorption of orange peel. However, SEM micrographs of modified orange peels revealed major changes in the structural properties of the peel after modification processes. Physical modification increased the surface area of orange peel, providing more surface area for the adsorption of contaminants, while chemical modifications of orange peel increased the number of binding sites in the orange peel. Furthermore, the heating of orange peel during thermal modification releases gases within the pore spaces, resulting in highly porous structures and a subsequent adsorption increase; however, the porosity and surface area of orange peel decreases when heated at temperatures above 500 °C. Finally, incorporating chemicals into thermal treatments improves the structural properties of orange peel and enhances the adsorption capacities of the peel. In addition, the adsorption capacities of modified orange peels reviewed in this study revealed that orange peels treated with chemicals recorded the highest adsorption capacities. Treating orange peels with a base, such as sodium hydroxide, converts methyl esters to carboxylates, and increases the number of binding sites and biosorption capacities of orange peel.

### Recommendation

The study observed that most of the existing literature on orange peel biosorption are studies on the adsorption of heavy metals, and there are a few on the adsorption of dyes. However, limited studies exist on the adsorption of oil contaminants by orange peel. Therefore, the researchers recommend that future studies should focus on orange peel adsorption of oil contaminants.

## Figures and Tables

**Figure 1 materials-16-01092-f001:**
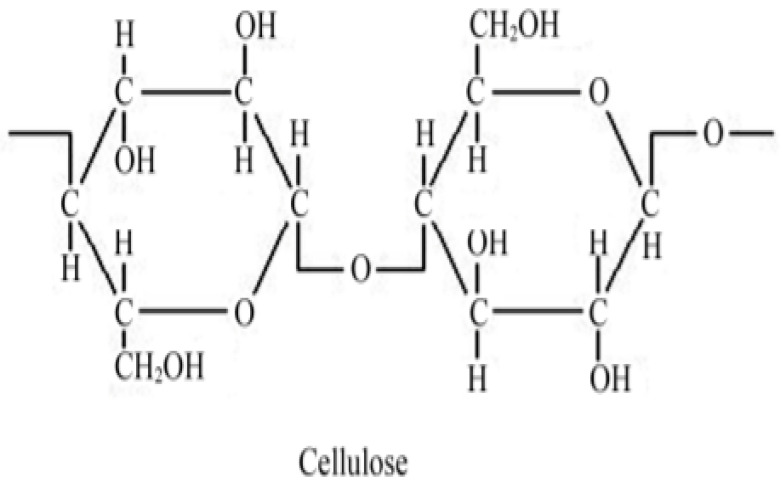
Structure of cellulose [79].

**Figure 2 materials-16-01092-f002:**
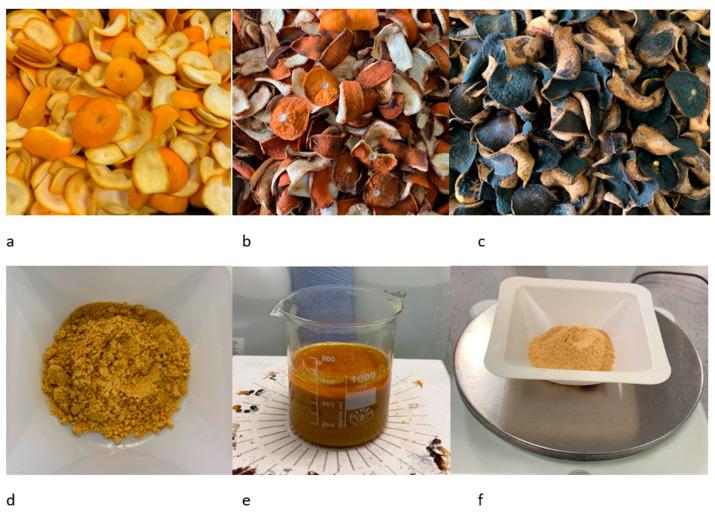
Different stages of orange peel adsorbent preparation: (**a**) fresh orange peels, (**b**) orange peels dried for 24 h at 75 °C, (**c**) orange peels dried for 72 h at 75 °C, (**d**) dried orange peel powder, (**e**) orange peel soaked in NaOH and ethanol solution, (**f**) chemically modified orange peel.

**Figure 3 materials-16-01092-f003:**
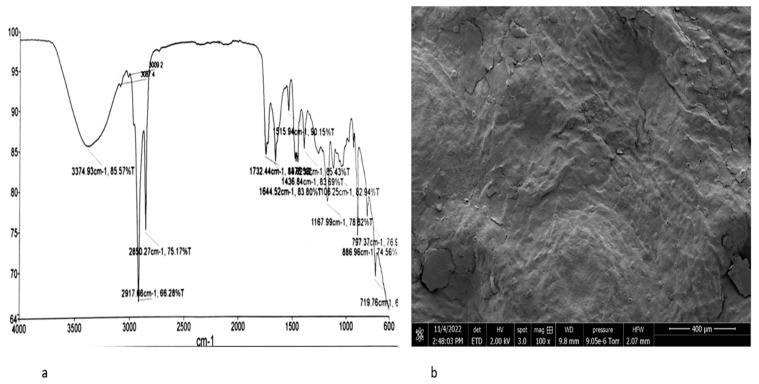
FTIR spectrum (**a**) and SEM image (**b**) of raw orange peel (this study).

**Figure 4 materials-16-01092-f004:**
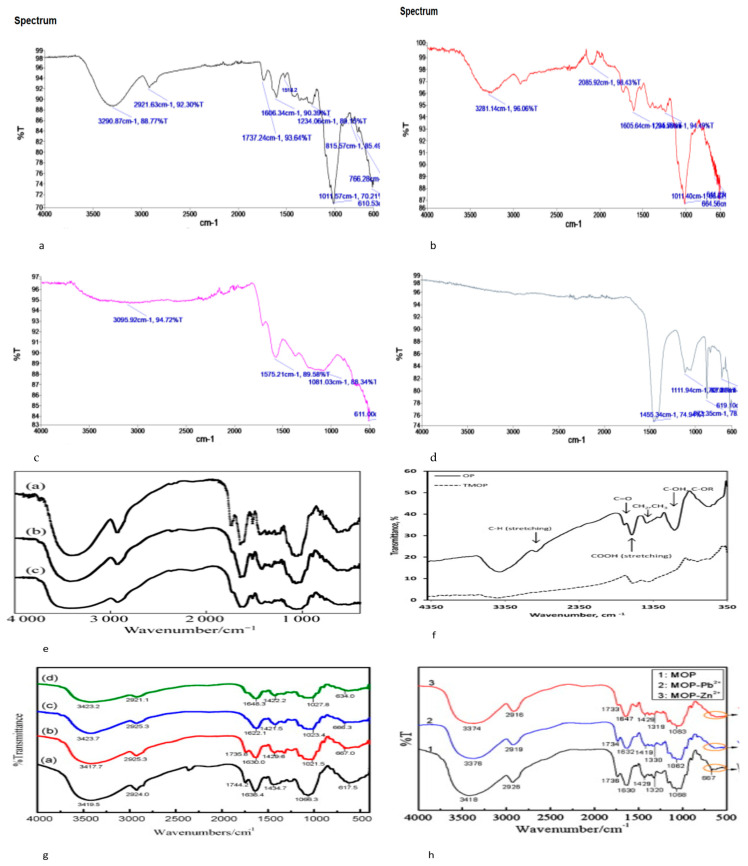
(**a**) FTIR spectra of physically modified orange peel, (**b**) chemically modified OP, (**c**) thermally modified OP at 300 °C, (**d**) thermally modified OP at 500 °C (this study), (**e**) OP, chemically modified OP, chemically modified OP after adsorption [88], (**f**) OP, thermally modified OP [66], (**g**) FTIR spectra of OP, MOP, and MOP after adsorption [89], (**h**) FTIR spectra of OP, MOP, and MOP after adsorption [90].

**Figure 5 materials-16-01092-f005:**
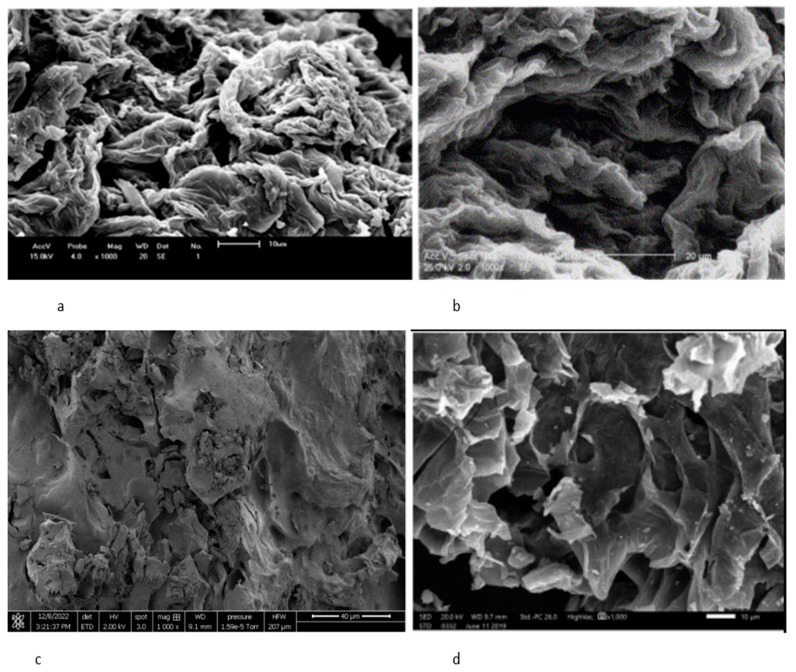
(**a**) SEM images of physically modified orange peel, (**b**) physically modified OP after adsorption [74], (**c**) thermally modified orange peel at 300 °C (this study), (**d**) thermally modified orange peel at 300 °C [13].

**Figure 6 materials-16-01092-f006:**
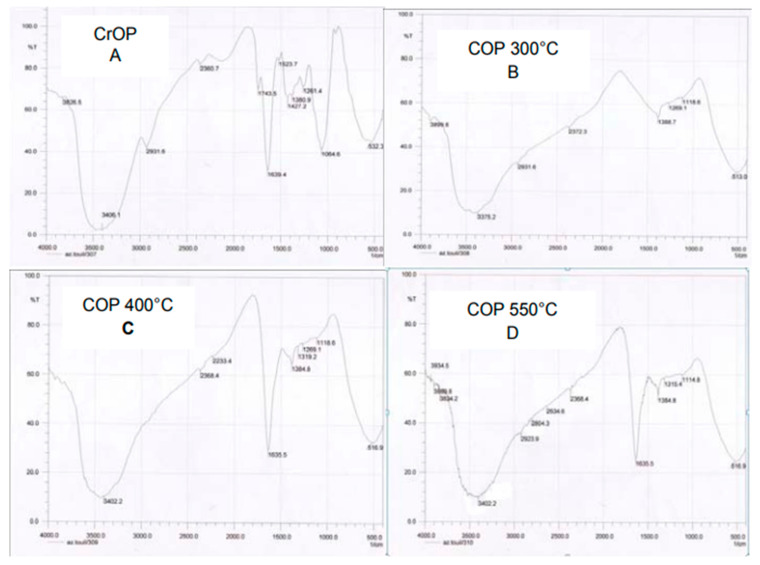
(**A**) FTIR of raw orange peel, (**B**) Calcined at 300 °C, (**C**) 400 °C, (**D**) 550 °C, (**e**–**g**) Effects of time on adsorption capacities of OP calcined at different temperatures [101].

**Figure 7 materials-16-01092-f007:**
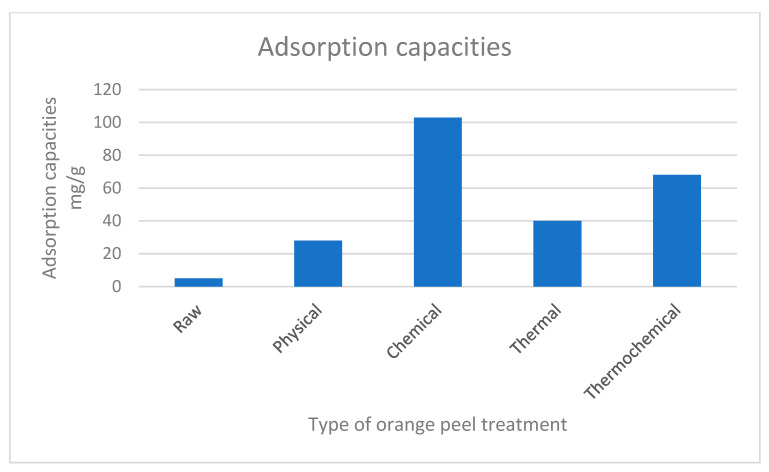
Adsorption capacities of modified orange peels.

**Table 2 materials-16-01092-t002:** Water contact angle of some sorption lignocellulose materials.

S/No	Adsorbent	Water Contact Angle	Sorbate	Maximum Adsorption Capacity mg/g	References
1	Kapok fibre	138.6–151.2°	Diesel Soybean	38.1 49.1	[61]
2	Cotton fibre	100°	Vegetable oil	30	[70]
3	Wool fibre	155°	Diesel Vegetable oil	10.6 14.5	[51]
4	Lotus fibre	161°	Dye	24.14	[71]
5	Banana peel	110°	Oil	5	[72]
6	Orange peel	0°	Dye	5	[14]

**Table 3 materials-16-01092-t003:** Elemental composition of orange peel.

Chemical Composition	Mass% [76]	Mass% [77]	Mass% [78]	Mass% This Study
Carbon	49.59	44.5	47.0	48.67
Hydrogen	6.95	6.1	6.0	-
Oxygen	39.7	47.3	44.71	36.46
Na	-	-	-	4.44
Nitrogen	0.66	1.5	1.3	-
Potassium	-	-	-	0.95
Calcium	-	-	-	1.08
Sulphur	0.06	0.4	0.09	-
Chloride	0.001	-	0.001	8.39
Ash	3.05	4.0	-	-
Water	2.73	-	-	-

**Table 4 materials-16-01092-t004:** Summary of adsorption studies with orange peel.

References	Author	Adsorbent State	Adsorption Capacities (mg/g)	Adsorbate
Application of chemically modified orange peels for removal of copper (II) from aqueous solutions	[90]	Thermochemical	-	Heavy metal –copper (II)
Effect of calcination on orange peels characteristics: Application of an Industrial Dye Adsorption	[94]	Thermal -calcination	9.74	Dye –methylene blue (MB)
Oil spill sorption capacity of raw and thermally modified orange peel waste	[95]	Raw Thermal	30–50	Oil
Potential application of orange peel as an eco-friendly adsorbent for textile dyeing effluents	[88]	Dried (crushed)	-	Dye
Effect of temperature on the adsorption of methylene blue dye onto sulfuric acid–treated orange peel	[97]	Thermochemical		Dye
Equilibrium and thermodynamic studies of Cd (II) biosorption by chemically modified orange peel	[74]	Chemical		Heavy metal
Equilibrium study of dried orange peel for its efficiency in removal of cupric ions from water	[102]	Dried (crushed)	33.99 mg/g	Heavy metal –copper ion
Functionalised adsorbents prepared from fruit peels: Equilibrium, kinetics, and thermodynamic studies for copper adsorption in aqueous solution	[103]	Physical—Chemical	163 mg/g	Heavy metal –Cu (II)
Preliminary study of oil removal using hybrid peel waste: Musa Balbisiana and Citrus Sinensis	[10]	Chemical (NAOH)	-	Oil (Light and Heavy)
Removal of Direct N Blue—106 from artificial textile dye effluent using activated carbon from orange peel: Adsorption Isotherm and Kinetic studies	[17]	Chemical	107.53 mg/g	Dye
Treatment of artificial textile dye effluent containing Direct Yellow 12 by orange peel carbon	[93]	Chemical—Thermal	75.76 mg/g	Dye
Sorption of Iron, Manganese, and Copper from Aqueous solution using orange peel: Optimization, Isothermic, Kinetic, and Thermodynamic studies	[92]	Chemical	15 mg/g	Heavy metal
Adsorption of Remazol Brilliant Blue on an Orange Peel Adsorbent	[74]	Dried (crushed)	9.7 mg/g	Dye
Simple and green fabrication of recyclable magnetic highly hydrophobic sorbents derived from waste orange peels for removal of oil and organic solvents from water surface	[14]	Chemical	54.20 mg/g	Oil
Investigation of aqueous Cr (VI) adsorption characteristics of orange peels powder	[102]	Chemical	4.69 mg/g	Heavy metal
Application of orange peel Xanthate for the adsorption of Pb^2+^ from aqueous solutions	[94]	Chemical	204.50 mg/g	Heavy metal
Adsorption of Pb^2+^ and Zn^2+^ from aqueous solutions by sulphured orange peel	[90]	Chemical	160 mg/g 80 mg/g	Heavy metal Pb^2+^ Zn^2+^
Removal of Heavy Metal Ions from Aqueous Solutions by Adsorption using modified orange peel as adsorbent	[96]	Chemical	14.1 mg/g–45.29 mg/g	Heavy metal
Adsorption of CU^2+^ and Cd^2+^ from aqueous solution by Mercapto—acetic acid modified orange peel	[89]	Chemical	70.67 mg/g 136.05 mg/g	Heavy metal Cu^2+^ Cd^2+^
Adsorption study of copper (II) by chemically modified orange peel	[95]	Chemical	289.0 mg/g	Heavy metal Copper (II)
Kinetic and thermodynamic studies on biosorption of Cu (II) by chemically modified orange peel	[104]	Chemical	72.73 mg/g	Copper (II)
Biosorption of heavy metals from aqueous solutions by chemically modified orange peel	[105]	Chemical	162.6	Heavy metals
Characterization of adsorptive capacity and mechanisms on adsorption of copper, lead, and zinc by modified orange peel	[106]	Chemical	70.73 209.8 56.18	Heavy metal Cu^2+^ Pb^2+^ Zn^2+^
Copper biosorption from aqueous solutions by sour orange residue	[107]	Chemical	21.7	Heavy metal
Adsorption of Remazol Brilliant Blue on an orange peel adsorbent	[74]	Dried (crushed)	9.7 mg/g	Dye
Enhanced Cu (II) adsorption by orange peel modified with sodium hydroxide	[88]	Chemical	50.25 mg/g	Heavy metal Cu (II)
Use of chemical modification to determine the binding of Cd (II), Zn (II), and Cr (II) ions by orange waste	[91]	Chemical	41.59 mg/g 32.04 mg/g 40.35 mg/g	Heavy metal Cd^2+^ Zn^2+^ Cr^2+^
Application of orange peel waste in the production of solid biofuels and biosorbents	[77]	Thermal (pyrolysis)	-	Heavy metal
Adsorption/desorption of Cd (II), Cu (II), and Pb (II) using chemically modified orange peel: Equilibrium and Kinetic studies	[108]	Chemical	13.7 mg/g 15.27 mg/g 73.53 mg/g	Heavy metals Cd Cu Pb
Removal of dyes from coloured textile wastewater by orange peel adsorbent: Equilibrium and kinetic studies	[16]	Physical (dried)	10.72 mg/g and 21.05 mg/g	Dyes
Enhanced removal of reactive navy-blue dye using powdered orange waste	[109]	Physical (dried)	30.28 mg/g	Dye
Arsenic(V) biosorption by charred orange peel in aqueous environments	[19]	Thermochemical	60.9 mg/g	Dye
Characteristic and biosorption capacities of orange peels biosorbents for removal of ammonia and nitrate from contaminated water	[110]	Chemical	100%	Ammonia Nitrate
Adsorptive Removal of 4-Nitrophenol from Aqueous Solution by Activated Carbon Prepared from Waste Orange Peels	[111]	Chemical	73.35	4-Nitrophenol
Waste to resource recovery: mesoporous adsorbent from orange peel for the removal of trypan blue dye from aqueous solution	[112]	Chemical	97.10%	Dye
Characterization of banana and orange peels: biosorption mechanism	[39]	Physical	-	-
Characterization and application of orange peel as an adsorbent for cationic dye removal from aqueous solution	[113]	Physical	3.96	Dye
Preparation and evaluation of orange peel cellulose adsorbents for effective removal of cadmium, zinc, cobalt, and nickel	[114]	Chemical	130%	Metals
Biosorbents prepared from orange peels using Instant Controlled Pressure Drop for Cu (II) and phenol removal	[115]	Thermal	32.51 106.91	Metals
Activated carbon derived from waste orange and lemon peels for the adsorption of methyl orange and methylene blue dyes from wastewater	[116]	Chemical	33 38	Dye

## Data Availability

Not applicable.

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
