# Peer review of "Modified Orange Peel Waste as a Sustainable Material for Adsorption of Contaminants"

_materials, 2023, doi:10.3390/ma16031092_

Round 1

Reviewer 1 Report

In this paper, the author used modified orange peel waste for adsorption of contaminants. It is an interesting work. However, the paper in current form is lack of standardization as an scientific paper. Please revise the paper carefully and double check the errors.

1. There is no figure number in Figure 2, 4. 

2. There is no scale bare for SEM images.

3. All the figures (especially FTIR patterns) are not clear enough.

4. Where is Figure 3?

5. There are too many references in the manuscript. Please put the unnecessary references in the supporting information.

6. The adsorption mechanism is not clear. Please provide the FTIR patterns of the modified orange peels after adsorption.

7. What is reference No. 122?

Reviewer 2 Report

Language must be improved. Many typo errors. Resolution of images must be improved. Heterogeneity in reference section, must strictly follow single format. Many important points as comments must be addressed. Needs moderate revision, after which paper can be considered for publication.

Reviewer 3 Report

comments on the manuscript

·         Rewrite the references in order, and the first reference is numbered with 1, and it is the first reference in the list of references and so on.

·         Scheck the data mention in abstract “3 g/g  to 5 g/g”. I think it 3 mg/g to 5 mg/g

·         Write the abbreviation of time hr. in all text

·         Write ᵒC not ᵒc for the Celsius abbreviation

·         Check the reference format

·         Line 37: add reference to this statement

·         Table 1: write the missing Adsorption capacity

·         Title 1.1 and 1.2 should not be the subtitle after introduction. Add main title 2 to include these two subtitles

·         Line 73: correct the font

·         Table 2: complete the missing data, add orange peel in the table and change its title with anther suitable title

·         Line 96 :put “ and” before ash

·         Line 161: “Figures 3a and b show FTIR and SEM images of orange peel” this data already mention above

·         The FTIR figure: the data on the figure not clear at all

·         Line 162: this Figure 3 not 4

·         The “OP” did not define in the manuscript

·         “2.2 Physical modification” Rewriting it in an orderly and clear manner for the data, and the images are defined in the publication and interpreted in order according to their appearance in the publication

·         Line 270, 271, and 274 :Write the unite of the peak of FTIR “ cm-1”

·         Table 4 This table was written very carelessly. Rewriting and arranging the information it contains and completing the missing data . mention the dye name in the adsorbate column

·         Figure 7: change the X axis to be “Type of orange peel treatment”

Round 2

Reviewer 1 Report

Before publication, it is suggested to put all the tables with too many references in supporting information to make the paper concise.

Reviewer 3 Report

Thank you for the corrections you have made in the text, but there are references in the same field  (https://doi.org/10.3390/molecules27061831), (https://doi.org/10.1016/j.fuel.2022.124288) and (https://www.deswater.com/DWT_abstracts/vol_218/218_2021_423.pdf) that contain helpful information that can be referred to and benefited from.